# Sophorolipid Suppresses LPS-Induced Inflammation in RAW264.7 Cells through the NF-κB Signaling Pathway

**DOI:** 10.3390/molecules27155037

**Published:** 2022-08-08

**Authors:** Ruiqi Xu, Ling Ma, Timson Chen, Jing Wang

**Affiliations:** 1Key Laboratory of Synthetic and Biological Colloids, Ministry of Education, School of Chemical and Material Engineering, Jiangnan University, Wuxi 214122, China; 2Adolph Innovation Laboratory, Guangzhou Degu Personal Care Products Co., Ltd., Guangzhou 510000, China

**Keywords:** sophorolipid, biosurfactants, LPS, anti-inflammatory, NF-κB pathway

## Abstract

Objectives: Biosurfactants with anti-inflammatory activity may alleviate skin irritation caused by synthetic surfactants in cleaning products. Sophorolipid (SL) is a promising alternative to synthetic surfactants. However, there are few reports on the anti-inflammatory activity of SL and the underlying mechanism. The purpose of this work is to verify that lipopolysaccharide (LPS)-induced inflammation could be inhibited through targeting the pathway of nuclear factor-κB (NF-κB) in RAW264.7 cells. Methods: The influence of SL on cytokine release was investigated by LPS-induced RAW264.7 cells using ELISA. The quantification of the protein expression of corresponding molecular markers was realized by Western blot analysis. Flow cytometry was employed to determine the levels of Ca^2+^ and reactive oxygen species (ROS). The relative expression of inducible nitric oxide synthase (INOS) and cyclooxygenase-2 (COX-2) was determined by RT-PCR. An immunofluorescence assay and confocal microscope were used to observe the NF-κB/p65 translocation from the cytoplasm into the nucleus. The likely targets of SL were predicted by molecular docking analysis. Results: SL showed anti-inflammatory activity and reduced the release of inflammatory cytokines including interleukin-6 (IL-6), tumor necrosis factor-α (TNF-α), and nitric oxide (NO). The experimental results show that SL suppressed the Ca^2+^ and ROS levels influx in the LPS-induced RAW264.7 cells and alleviated the LPS-induced expression of iNOS and COX-2, the LPS-induced translocation of NF-κB (p65) from the cytoplasm into the nucleus, and the expression of phosphorylated proteins such as p65 and IκBα. Furthermore, molecular docking analysis showed that SL may inhibit inflammatory signaling by competing with LPS to bind TLR4/MD-2 through hydrophobic interactions and by inhibiting IKKβ activation through the hydrogen bonding and hydrophobic interactions. Conclusion: This study demonstrated that SL exerted anti-inflammatory activity via the pathway of NF-κB in RAW264.7 cells.

## 1. Introduction

Surfactants are applied in the manufacture of household chemicals, cosmetics, and many other commercial products, making them the substances that come into contact with the skin most frequently [1,2]. However, surfactants, especially synthetic surfactants, are among the most important factors that may cause skin irritations [3,4]. For example, sodium dodecyl sulfate, a surfactant commonly used in cleaning products, is a typical skin irritant. Skin irritation is considered to be a locally reversible inflammatory response. Surfactants can break the skin barrier and penetrate into the skin, and then they can interact with proteins and cellular cytoplasm to trigger immune activation and the generation of pro-inflammatory cytokines, leading to skin inflammatory responses such as contact dermatitis, localized erythema, or pruritus [1].

Biosurfactants produced by microorganisms are usually more biodegradable and eco-friendly and less irritating than synthetic surfactants [2,5,6,7]. The combination of biosurfactants and synthetic surfactants in the formulation can improve the skin gentleness of cleaning products. More importantly, some of the biosurfactants have anti-inflammatory activities [6,8]. As reported by Zhang et al., surfactin, as one kind of biosurfactant, had an anti-inflammatory influence and could significantly suppress LPS-induced RAW264.7 macrophages from expressing interferon-γ (IFN-γ), IL-6, NO, and iNOS. Morita et al. found that biosurfactant mannosylerythritol lipids had an anti-inflammatory effect that prevented the mast cells from secreting inflammatory mediators [8]. These biosurfactants with anti-inflammatory activity may alleviate the skin irritation caused by other synthetic surfactants in cleaning products.

Sophorolipid (SL), as one kind of biosurfactant, includes a fatty acid as the aglycone and a sophorose residue (a disaccharide that links two glucose residues through the β-1,2′ bond). SL can lower the surface tension of water to as low as 34.2 mN/m, and the mean CMC of SL is smaller than that of classical head and tail ionic surfactants and comparable to that of non-ionic surfactants [9]. SL not only displays a surface-lowering property but also exhibits anti-inflammatory, antiviral, antimicrobial, and other biological activities [10,11,12]. Moreover, the high yield of SL production by non-pathogenic yeasts makes it commercially attractive and provides a promising alternative for the synthesis of surfactants. In the animal model studies conducted by Hardin et al., SL exhibited an anti-inflammatory effect [13]. Hagler et al. reported that SL could work as a potential therapeutic compound and anti-inflammatory agent due to its ability to downregulate the immunoglobulin-E (IgE) coding genes in U266 cells [14]. Nevertheless, the mechanisms underlying the anti-inflammatory properties of SL have not been revealed, especially in macrophages. In this study, the anti-inflammatory effects of SL were investigated in the macrophages of LPS-induced RAW264.7. SL was found to reduce the levels of proinflammatory cytokines and suppress the activation of the NF-κB pathway in RAW264.7 cells in this work. It provides the anti-inflammatory mechanisms of SL, which will be the fundamental basis for the application of SL in ultra-mild cleaning products.

## 2. Materials and Methods

### 2.1. Fermented Production of SL

The SL used in this study was obtained by the fermentation of a non-pathogenic Candida bombicola (ATCC 22214 from SHBCC, Shanghai, China). First, C. bombicola cells were incubated in 50 mL of yeast extract–malt extract broth (the mass ratio of yeast extract, peptone, and glucose was 1:2:2, and their total mass was 50 g per liter.) oscillating in an incubator (220 rpm, 30 °C, and 48 h). Then, a jar fermenter (Bailun bio, Shanghai, China) with a total volume of 5.0 L and a working volume of 2.0 L was used for the fed-batch cultivation. A culture suspension with a concentration of 5% (*v*/*v*) was selected to be the inoculum. The fermentation medium for the production has the composition of glucose (100 g/L), rapeseed oil (50 g/L), yeast extract (5 g/L), KH_2_PO_4_ (1 g/L), peptone (0.7 g/L), MgSO_4_·7H_2_O (0.5 g/L), NaCl (0.1 g/L), and CaCl_2_·2H_2_O (0.1 g/L). The culture conditions were as follows: agitation, 550 rpm; temperature, 30 °C; working volume, 2 L; culture time, 5–6 days; aeration rate, 2–6 L/min; and pH value, 3.8, which was adjusted by automatically adding 6 mol/L NaOH solution when it decreased in the fermentation process. When the concentration of glucose decreased to <30 g/L in this process, glucose was added to keep its concentration no lower than 30 g/L, and no additional glucose was added during the last 48 h of the fermentation process. In addition, the concentration of rapeseed oil was maintained above 20 g/L to provide abundant hydrophobic substrates. When the glucose was exhausted, the fermentation was finished.

### 2.2. Extraction and Purification of SLs

Since natural SLs are insoluble in water in the fermentation broth with pH = 3.8, it will separate out in the form of a brown, viscous, oily precipitate at the bottom of the reactor after stopping the aeration and stirring, while the remaining low-density rapeseed oil will be separated from the water phase by floating to the top of the reactor. After collecting the precipitate, the pH was adjusted to neutral by using 6 mol/L NaOH solution to increase the solubility of SLs in the water. The cells were removed using high-speed centrifugation (12,000 rpm, 9 min). The supernatant was collected and extracted with anhydrous ethanol that was twice its volume, and then the impurities from the solution were precipitated using high-speed centrifugation (8000 rpm, 10 min). After the ethanol was evaporated, a certain concentration of purified SLs was obtained.

### 2.3. Characterization of SLs

Liquid chromatography (LC) equipment: MALDI SYNAPT Q-TOF MS (Waters, Milford, MA, USA) with a diode array detector; column, BEH C18 2.1 mm × 150 mm, 1.7 μm; column temperature, 30 °C; injection volume, 5 µL; detection wavelength, UV 207 nm. A formic acid aqueous solution of 0.5 wt% and acetonitrile were used as the mobile phases A and B, respectively. Gradient elution was performed using 50% B at 0 min, 50% B at 5 min, 60% B at 30 min, 100% B at 50 min, and 100% B at 60 min with a flow rate of 0.3 mL/min.

Mass spectrometry (MS) equipment: electrospray ionization in positive ion mode; voltage of capillary, 3.5 kV; range of scanning, *m*/*z* 200–900; pressure of spray gas, 344.5 kPa; drying gas, high-purity N_2_ with a gas temperature of 350 °C and a gas flow of 11.0 L/min; voltage of cone, 65 V; voltage of lysis, 100 V; reference solution, *m*/*z* 121.05, 922.01.

### 2.4. Cell Culture

The RAW264.7 cells used in this study were provided by BeNa Culture Collection (BNCC337875, Shanghai, China). Dulbecco’s modified Eagle medium (high glucose) (Gibco, Carlsbad, CA, USA) was used to culture the cells at 37 °C in a humidified atmosphere with 5% CO_2_, and the concentration of fetal bovine serum (Gibco, Carlsbad, CA, USA) was 10% (*v*/*v*). The cells were passaged by gently blowing with PBS, centrifuged at 1000 rpm for 5 min, resuspended with the complete culture medium, and then transferred to a new culture flask.

### 2.5. MTT Assay for Cell Viability

The RAW264.7 cells cultured in a 96-well plate (5 × 10^4^ cells/mL, 100 μL per well) were incubated for the final concentrations of 1, 3, 6, 10, 12.5, 20, 25, 50, and 100 μg/mL with and without the SLs solution for 24 h. Each concentration had four replicate wells. The models were cultured in 1 µg/mL LPS medium (ST1470, Beyotime Inc., Shanghai, China). The solution containing 0.5 mg/mL 3-(4,5-dimethylthiaxolone-2-yl)-2,5-diphenyl tetra-zoliumbromide (MTT) (Sigma-Aldrich Co., Saint Louis, MO, USA) of 100 μL was poured into each well, followed by another six hours of incubation at 37 °C. After that, the medium was carefully separated, and formazan crystals were solubilized by the addition of 100 μL of dimethyl sulfoxide (DMSO, Sinopharm Chemical Reagent Co., Beijing, China) into each well. An enzyme marker (Tecan Infinite 200Pro) was used to measure the absorbance at 490 nm. The cell viability was calculated according to the formula below:Cell viability (%) = (OD_T_/OD_B_) × 100%(1)
where OD_T_ and OD_B_ represent the average OD of the experimental group and the blank group, respectively.

### 2.6. Measurement of Cytokine and NO, TNF-α, and IL-6 Concentrations

The cells inoculated in 24-well plates were divided into the of groups of blank, LPS, and SL+LPS and incubated for 24 h. Then, the supernatant was carefully pipetted off. For the blank group, 500 µL of the DMEM solution was added. For the LPS group, 500 µL of the DMEM solution containing LPS (1 µg/mL) was added. For the SL+LPS group, the DMEM solution with different concentrations of SL (6, 12.5, 20 µg/mL) was pretreated for 2 h, and then the final concentration was kept to 1 µg/mL by adding LPS and incubated in the incubator for another 24 h. After that, the supernatants used for the assays of NO and cytokine were collected. For the measurement of NO, the above supernatant was mixed with an equal volume of Griess reagent (S0021S, Beyotime Inc., Shanghai, China) and then incubated for 10 min at room temperature. The concentration of nitrogen oxide was measured by oxygen demand (OD) analysis at 540 nm. NaNO_2_ (S0021S, Beyotime Inc., China) was used as a standard regent, and the enzyme-linked immunosorbent assay kits (Absin Bioscience Inc., Shanghai, China) were used to determine the concentration of cytokine for IL-6 and TNF-α.

### 2.7. Measurement of Intracellular ROS and Ca^2+^ Levels

The experimental grouping was the same as that in Section 2.6. After co-incubation for 8 h in 6-well plates, the culture medium was pipetted off and incubated for 20 min with DCFH-DA (10 μmol/mL, S0033S, Beyotime Inc., Shanghai, China) as a probe for ROS detection and for 1 h with Fluo-4AM (5 μg/mL, S1060, Beyotime Inc., Shanghai, China) as a probe for Ca^2+^ detection, respectively. After that, the supernatant incubation medium was discarded. After washing twice with 400 μL PBS, the cells were washed down and collected into flow tubes. Flow cytometry (BD FACS Arica III, Franklin Lake, NJ, USA) was used to determine the levels of intracellular ROS/Ca^2+^ production.

### 2.8. Measurement of the mRNA Expression of iNOS and COX-2

The experimental grouping was the same as that in Section 2.6. After co-incubation for 8 h in 6-well plates, the extraction of total RNA was performed with the Rneasy^®^ RNA extraction kit (Beyotime Inc., Shanghai, China). The cDNA was synthesized with a cDNA synthesis kit on ThermoCycler (Thermo Fisher Scientific, Waltham, MA, USA). The relative expression of the target genes was calculated using the 2^−ΔΔCt^ method, with GAPDH taken as the negative control to normalize the relative expression. The following primers were used in this study: GAPDH (F: 5′-GGCCTTCCGTGTTCCTACC-3′; R: 5′-TGCCTGCTTCACCACCTTC-3′); iNOS (F: 5′-AGCAACTACTGCTGGTGGTG-3′; R: 5′-TCTTCAGAGTCTGCCCATTG-3′); COX-2 (F: 5′-TGAGTACCGCAAACGCTTCTC-3′; R: 5′-TGGACGAGGTTTTTCCACCAG-3′).

### 2.9. Immunofluorescence

The cells were cultured in laser confocal dishes, and the experimental grouping was the same as that in Section 2.6. After the pretreatment of the SL+LPS group with different concentrations of SL for 24 h, LPS was added to make its final concentration be 1 µg/mL for one hour, which was the same as that for the LPS group. Then, the cells were washed once in PBS, fixed in the fixative for 15 min, washed three times, added to the immunostaining blocking solution, and blocked at room temperature for 1 h. Thereafter, the NF-κB (p65) primary antibody was incubated and washed three times, and the secondary antibody was incubated, washed, incubated with 4′,6-diamidino-2-phenylindole (DAPI) stain, washed, and sealed with blocker. At last, the slices were photographed by a laser confocal microscope (Leica TCS SP8, Vizsla, Germany). The antibodies and reagents used for immunofluorescence were from Shanghai Beyotime Biotechnology (Shanghai, China).

### 2.10. Western Blot Assay

The experimental grouping was the same as that in Section 2.6. After co-incubation in 6-well plates for 18 h, the supernatant cultures were pipetted off and the cells were collected. Then, the cell proteins were extracted and their concentrations were measured with a BCA kit (P0012, Beyotime Inc., Shanghai, China). The related proteins NF-κB p65 (p65), NF-κB p65 phosphorylated protein (P-p65), IκBα, and IκBα phosphorylated protein (P-IκBα) were then assayed by Western blotting, using GAPDH as an internal reference. All the antibodies were from Cell Signaling Technology, and they were applied after dilution. Finally, strip imaging was performed by the ChemiDOC XRS+ Gel Imaging System (BIO-RAD, Hercules, CA, USA).

### 2.11. Molecular Docking

For the docking analysis, the protein structures of TLR4/MD-2 (PDB ID: 5IJD) and IKKβ (PDB ID: 3RZF) were provided by Protein Data Bank (PDB, https://www.rcsb.org/, assessed on 7 June 2021), and the 3D structure of the SL (Compound ID: 117065237) was obtained from PubChem (https://pubchem.ncbi.nlm.nih.gov/, assessed on 7 June 2022). The deletion of water and ligand molecules from the protein structures, the hydrogenation, and the charge manipulation were handled with Discovery Studio software. The target protein and ligand were docked using Autodock Vina software with default parameters, and all the possible conformations were obtained. The estimated free energy of the binding was used to evaluate the results.

### 2.12. Statistical Analysis

The experimental data in this study were shown in the form of the mean ± standard error and analyzed by GraphPad. The contrast in the various groups was conducted with analysis of variance (ANOVA), which was then conducted by Tukey’s post-hoc test. The significant difference in statistics was considered to be *p* < 0.05.

## 3. Results

### 3.1. Characterization of SL

As described by Baccile et al. [10], SL is usually obtained by fermentation as a mixture of several congeners, and the yield ratio of acidic and lactone forms is approximately 20:80. LC-MS was used to investigate the composition ratio and structure of the SL formed from rapeseed oil as a hydrophobic carbon source. As can be seen from Figure 1, the peaks of the acidic-SLs were concentrated before 3 min, whereas the peaks of the lactonic-SLs appeared within 3 to 10 min. There is a significant difference between the peaks of these two configurations, which indicates that the purified SL was a mixture of both acidic and lactonic types, while the latter accounted for 76.42%.

The MS analysis results in Table 1 indicate that the SL used in this study is a mixture of monoacetylated and diacetylated products. The diacetylated lactonic SL with a peak time of 4.43 min and a molecular weight of 688 accounts for the largest proportion of the product, and the analysis of the specific mass spectra is shown in the Appendix A.

### 3.2. Effects of SL on Cell Viability

The results of the potential cytotoxicity of SL examined by the MTT assay in RAW264.7 cells indicate that the cell viability was greater than 90% in the absence or presence of LPS (1 µg/mL), suggesting that SL with a concentration of up to 20 μg/mL had no cytotoxicity on RAW264.7 cells. Therefore, SL with a concentration of 6, 12.5, and 20 μg/mL was used in the subsequent experiments which are shown in Figure 2.

### 3.3. SL Suppresses LPS-Induced Inflammation in RAW264.7 Cells

It is well known that the inflammatory mediators such as NO can effectively regulate inflammation, and the pro-inflammation cytokines such as IL-6 and TNF-α can aggravate the development of inflammation. The results of the Griess experiments show that the NO level was significantly raised in LPS-induced RAW264.7 cells. The inhibition of SL pretreatment on the release of NO depended on the concentration (Figure 3A). In addition, the ELISA results with LPS-induced RAW264.7 cells showed that the secretion levels of IL-6 and TNF-α were significantly decreased by SL pretreatment (Figure 3B,C). The above results indicate that LPS-induced inflammation can effectively be suppressed by SL in RAW264.7 cells.

### 3.4. SL Downregulated LPS-Induced Levels of Intracellular ROS and Ca^2+^

Generally, the inflammatory response leads to a large increase in reactive oxygen species (ROS) and is closely related to the change in intracellular Ca^2+^ concentration. Flow cytometry analysis showed that LPS induction could stimulate the production of ROS and the influx of Ca^2+^. As shown in Figure 4, SL significantly reversed the LPS-induced levels of ROS and intracellular Ca^2+^ in RAW264.7 cells, which depended on the concentration.

### 3.5. SL Alleviated the LPS-Induced Expression of iNOS and COX-2

The expression of inducible nitric oxide synthase (iNOS) and the production of cyclooxygenase (COX) are important proxies for the oxidative inflammatory state of cells. As shown in Figure 5, we found that iNOS and COX-2 were significantly up-regulated by LPS stimulation but greatly down-regulated by SL treatment, indicating a significant inhibitory effect of SL on the LPS-induced activation of the oxidative state.

### 3.6. SL May Exert Anti-Inflammatory Effects through the NF-κB Signaling Pathway

It is well known that NF-κB, as a major transcription regulator, can induce the transcription of pro-inflammatory cytokines, so the influence of SL in activating the NF-κB pathway was investigated. For the control group, Figure 6 shows that the red fluorescent NF-κB (p65) was primarily presented in the cytoplasm but not in the blue nucleus stained by DAPI. For the LPS-induced group, intensive red fluorescence could be observed in the nucleus, indicating that there was a translocation for NF-κB (p65) from the cytoplasm into the nucleus caused by LPS. By introducing SL pretreatment at a concentration of 12.5 μg/mL, the LPS-induced NF-κB (p65) nuclear transcription could be significantly inhibited.

Moreover, the levels and phosphorylation of p65 and IκBα cytoplasmic proteins were detected by Western blot analysis. Figure 7 indicates that LPS induction significantly upregulated the expression of phosphorylated p65 and IκBα (P-p65 and P-IκBα). The SL pretreatment prevented the phosphorylated p65 and IκBα from expressing, which depended on the dose. Notably, after the pretreatment with 6 μg/mL of SL, the ratios of P-p65/p65 and P-IκBα/IκBα in RAW264.7 cells were decreased by 65.72% and 68.46%, respectively. Therefore, the activation of the inflammatory NF-κB pathway can be suppressed by SL in LPS-induced RAW264.7 cells.

### 3.7. Molecular Docking Analysis of SL with TLR4/MD-2 and IKKβ

To better understand the inhibitory effect of SL on NF-κB, the diacetylated lactonic SL ligand was docked into the active pocket of the TLR4/MD-2 protein and the active site of the IKKβ protein. After docking, the chimeric model with the lowest binding energy level was selected. The results indicate that SL can bind to the MD-2 and IKKβ proteins with the binding free energy of −9.4 kcal/mol and −8.3 kcal/mol, respectively. SL mainly depends on hydrophobic interactions with TLR4/MD-2 (Figure 8A). SL interacts with IKKβ through both hydrogen bonding and hydrophobic interactions (Figure 8B). The amino acids of CYS, GLU, ASP, and GYL interact with SL-IKKβ through hydrogen bonds. The amino acids of ILE, GLU, PHE, VAL, ALG, and LEU interact with SL-TLR4/MD-2 and SL-IKKβ through hydrophobic interactions. This strongly suggests that when inflammation occurs, SL can bind to receptor proteins on the cell membrane and intramembrane kinases, thus preventing the further transmission of inflammatory signals.

## 4. Discussion

Synthetic surfactants in cosmetics and household products may interact with skin and cause skin irritation, which is related to inflammatory responses. The use of biosurfactants can improve the gentleness of skin because some biosurfactants are reported to have multiple biological activities, especially in inflammation and cancer, potentially alleviating the skin irritation caused by other synthetic surfactants in cleaning products [1,4,8]. In terms of the therapeutic approach, ideal compounds may have the ability to downregulate the pro-inflammatory cytokines and block the NF-κB. Sophorolipid (SL) is a promising alternative to synthetic surfactants in cosmetics and household products. Nevertheless, there are few studies on the underlying mechanism of SL and its anti-inflammatory activity.

In almost all tissues of the body, macrophages, as immune cells, are the direct effects and main responders of exotic substances [15]. There is the participation of macrophages in various stages of inflammation such as initiation, development, and resolution [15,16]. RAW264.7 macrophages are a common cell model to study the immunomodulatory activities of various compounds. Recently, RAW264.7 macrophages were also utilized in the skin inflammation models for predicting skin irritation potential. Therefore, in order to estimate the anti-inflammatory influence of SL, well-researched RAW264.7 macrophages were used, which further revealed the above-mentioned signaling pathway. This work showed that, in RAW264.7 cells, SL had an inhibition effect on the LPS-induced proinflammatory responses, which was probably due to the inactivated NF-κB pathway. As far as we know, this is the first time the anti-inflammatory influences of SL in macrophages such as RAW264.7 cells and its relationship with the signaling pathway of NF-κB have been reported.

RAW264.7 cells can secrete various inflammatory mediators and pro-inflammatory cytokines including NO, TNF-α, IL-6, etc. NO is the major mediator of the oxidative stress response, which can participate in and aggravate the inflammatory response. Although a small amount of NO can scavenge pathogens and supply protective measures for the organism in the early stage of inflammation, excessive NO release will react with superoxide anion to produce peroxynitrite, causing local tissue damage and further enhancing inflammation [16,17]. In this study, it is proved that LPS stimulation has raised the level of NO, and SL co-treatment has inhibited the production of NO in a dose-dependent manner. Tumor necrosis factor-α (TNF-α) is a classical inflammatory marker which exists at high levels in a variety of inflammatory diseases. It is the center of the inflammatory cascade, which can induce the production of various inflammatory factors, mediate the immune response, and eventually lead to cell apoptosis and tissue damage. Its expression level directly reflects the severity of inflammation. Interleukin-6 (IL-6) is a multifunctional cytokine with both pro-inflammatory and anti-inflammatory effects, and its mode of action is associated with the level in organisms. IL-6, with its concentration within the normal range, is able to promote the immune defense response in organisms. A high concentration of IL-6 will cause a series of inflammatory injuries, and it is also an important mediator of various acute and chronic inflammatory responses [16,18]. The inflammation intensity in RAW264.7 cells induced by LPS was represented by the TNF-α and IL-6 levels in this work. Figure 3 shows that SL can significantly suppress the excessive release of TNF-α and IL-6 in the concentration range of 6–20 μg/mL. These results indicate that SL can inhibit the generation of these proinflammatory cytokines.

ROS is also a main mediator in the cellular oxidative stress response, and excessive intracellular ROS may damage the mitochondria and stimulate the inflammatory response by reducing the mitochondrial membrane potential [17,19]. As a second messenger mediating various biological responses in cells, Ca^2+^ plays an important role in maintaining normal cellular physiological functions. When cellular stimulation occurs, Ca^2+^ flows into the cell through calcium channels in the cell membrane, resulting in a significant increase in the content of intracellular free Ca^2+^, which is not conducive to the homeostasis of organisms and is easy to conduct stimulation, further exacerbating inflammation. Therefore, both ROS and Ca^2+^ play a vital role in maintaining the proliferation and differentiation of cells and in regulating apoptosis [20,21]. These results suggest that SL can reverse the generation of ROS and the influx of Ca^2+^ in LPS-induced RAW264.7 cells.

iNOS is an NO synthase whose expression directly determines NO secretion and is an important indicator for the detection of inflammatory responses. The regulation of inducible nitric oxide synthase expression is considered to be an important tool in the treatment of inflammatory diseases. COX has three isozymes: COX-1, COX-2, and COX-3, of which COX-2 is the inducible enzyme, a key enzyme that catalyzes the conversion of arachidonic acid to prostaglandins, and is very closely related to inflammation [18,22]. Thus, inflammation can be effectively controlled by reducing or inhibiting the activation of COX-2. The expression of iNOS and COX-2 in the groups pretreated with SL in this study was significantly lower than that in the model group, indicating that SL can exert anti-inflammatory effects by inhibiting the synthesis of iNOS and reducing the release of NO while down-regulating the gene expression of COX-2.

NF-κB, as the nuclear transcription factor, is an important transcriptional regulator in cells and performs a key function during the inflammatory response [23] It regulates the expression of cytosolic inflammatory factors, chemokines, and many other inflammatory mediators. NF-kB is present in the cytoplasm by binding with IκBα in living organisms. When the cell is induced by LPS, IκBα is phosphorylated by upstream kinases such as IKKβ and degraded by the ubiquitin–proteasome pathway, and, subsequently, the release of various inflammatory chemokines and cytokines is regulated by the translocation of NF-κB (p65) into the nucleus [17,23] Therefore, NF-κB transduction is a vital procedure in the development of inflammation. In this study, the results in Figure 6 indicate that SL pretreatment could significantly prevent NF-κB (p65) in the cytoplasm from translocating into the nucleus.

The activation of NF-κB causes the protein phosphorylation of IκBα and p65, which further contributes to the release of IL-6, TNF-α, and other inflammatory cytokines and exacerbates the inflammatory response. In this study, Western blotting analysis suggests that the phosphorylation of IκBα and p65 is also inhibited. In summary, as schematically illustrated in Figure 9, the translocation of NF-κB from the cytoplasm into the nucleus (p65) and the expression of the phosphorylated proteins of IκBα and p65 can both be inhibited by SL, suggesting that SL exhibits anti-inflammatory activity via the NF-κB pathway.

To investigate the specific targets of SL as supporting evidence of inhibiting NF-κB, molecular docking was used to analyze the interactions between SL and the molecular structures of TLR4/MD-2 and IKKβ, respectively. Toll-like receptors 4 (TLR4) are cell membrane surface receptors that usually bind with myeloid differentiation protein-2 (MD-2) to form a dimer and participate in the transduction of LPS recognition by cells, and the recognition site is located in the hydrophobic cavity of MD-2 [24,25] When LPS binds to TLR4/MD-2, it can promote the activation of kinases in this pathway and induce an inflammatory immune response. Molecular docking analysis is a promising tool in drug discovery and structural molecular biology by which the interactions between ligands and proteins can be analyzed at the atomic level. The results show that SL has a high binding affinity for Toll-like receptor complexes on cell membranes as well as intramembrane kinases.

## 5. Conclusions

Sophorolipid, as a kind of biosurfactant, was found to suppress lipopolysaccharide-induced inflammation in macrophages by significantly reducing pro-inflammatory mediators such as TNF-α, ROS, Ca^2+^, NO, iNOS, COX-2, and IL-6. In addition, anti-inflammatory mechanisms mediated by sophorolipid revealed that sophorolipid inhibited the NF-κB signaling pathway with possible targets of TLR4/MD-2 and IKKβ. This study shows that sophorolipid has the potential to alleviate skin inflammation among people and can be used to partially replace chemical-based surfactants in daily cleansing products to reduce the skin irritation caused by them in the future.

## Figures and Tables

**Figure 1 molecules-27-05037-f001:**
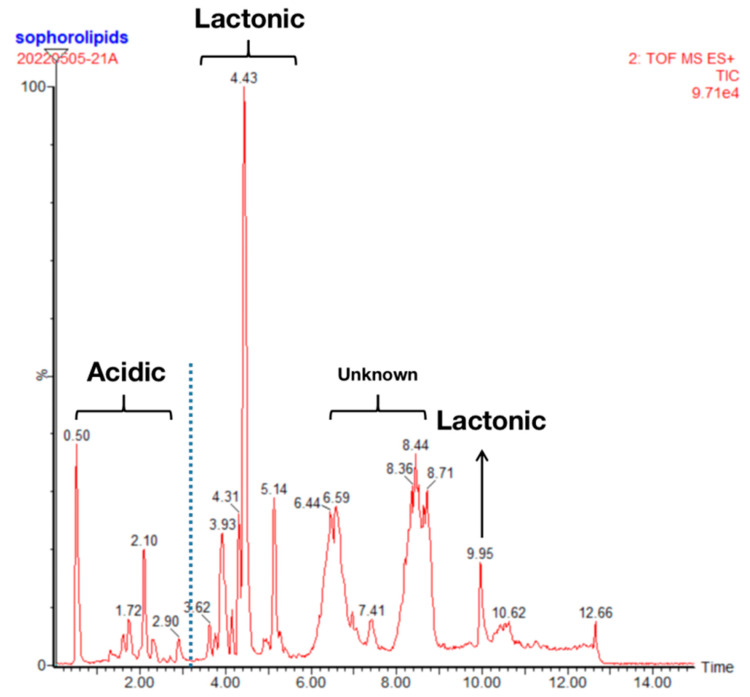
HPLC chromatogram of SL.

**Figure 2 molecules-27-05037-f002:**
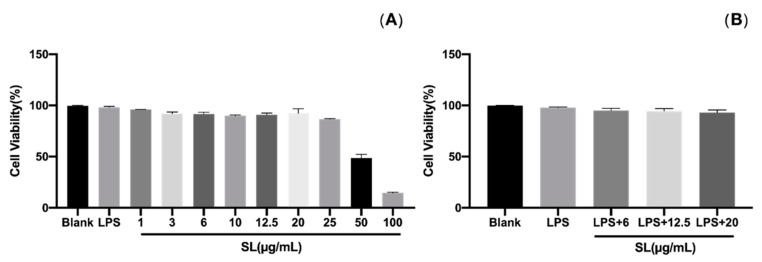
Effect of SLs on the viability of RAW264.7 cells. Pretreatment with or without LPS (1 µg/mL) on cells for 1 h; then, cells were treated with different concentrations of SLs for 24 h. (**A**) Cytotoxicity effect of SLs and LPS on RAW 264.7 cells. (**B**) Cytotoxicity effect of SLs on LPS-induced cell viability in RAW 264.7 cells. The results are presented as the mean ± standard error of three independent tests in triplicate.

**Figure 3 molecules-27-05037-f003:**
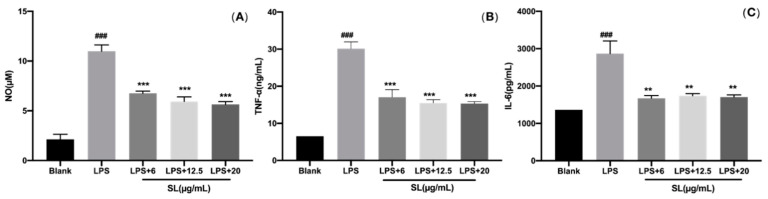
Effect of SLs on the LPS-induced inflammation in RAW264.7 cells. Cells were incubated with LPS (1 μg/mL) with or without SL for 24 h. (**A**) Secretions of NO were determined using a NO detection kit; (**B**) secretions of TNF-α were measured by a TNF-α ELISA kit; (**C**) secretions of IL-6 were measured by an IL-6 ELISA kit. The results are presented as the mean ± standard error (n = 3), and a p-value less than 0.05 was expressed as statistically significant (###, *p* < 0.001 vs. blank group; **, ***, *p* < 0.01, 0.001 vs. LPS treatment group).

**Figure 4 molecules-27-05037-f004:**
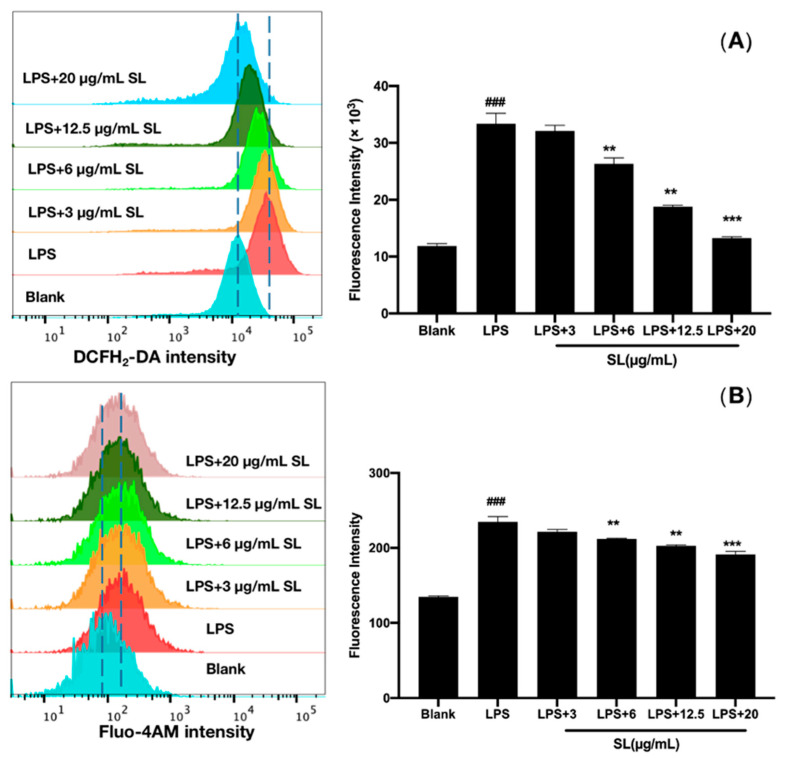
Effect of SLs on the LPS-induced levels of ROS and Ca^2+^ in RAW264.7 cells. Cells were incubated with 1 μg/mL of LPS with or without SL for 8 h. Flow cytometry was employed to determine the levels of (**A**) Ca^2+^ and (**B**) ROS. The results are presented as the mean ± standard error (n = 3), and a *p*-value less than 0.05 was expressed as statistically significant (###, *p* < 0.001 vs. blank group; **, ***, *p* < 0.01, 0.001 vs. LPS treatment group).

**Figure 5 molecules-27-05037-f005:**
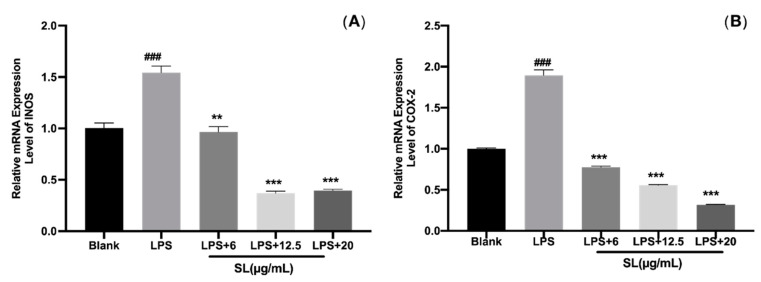
Effect of SLs on the LPS-induced expressions of iNOS and COX-2 in RAW264.7 cells. Cells were incubated with 1 μg/mL of LPS with or without SL for 8 h. The mRNA expressions of (**A**) iNOS and (**B**) COX-2 were analyzed by RT-PCR. The results are presented as the mean ± standard error (n = 3), and a *p*-value less than 0.05 was expressed as statistically significant (###, *p* < 0.001 vs. blank group; **, ***, *p* < 0.01, 0.001 vs. LPS treatment group).

**Figure 6 molecules-27-05037-f006:**
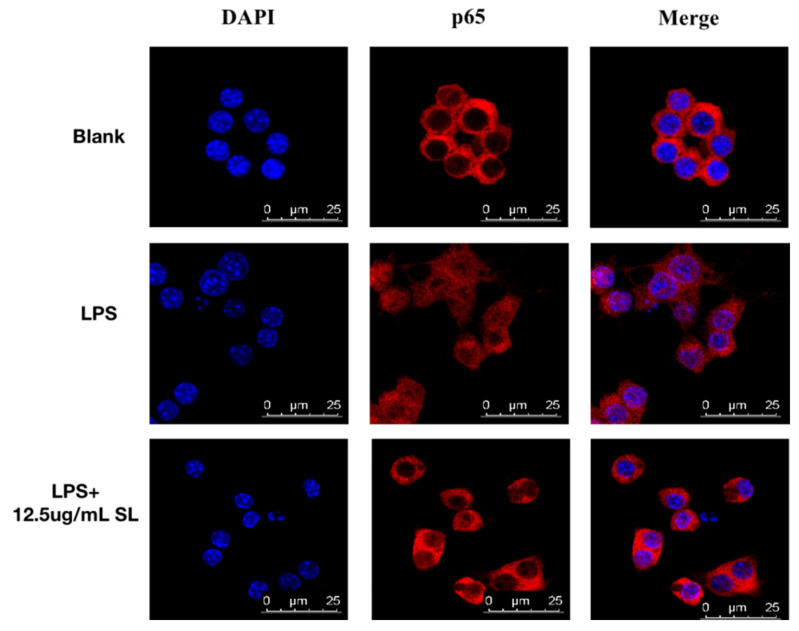
Effect of SL on nuclear translocation of NF-κB p65 in LPS-induced RAW 264.7 cells. Localization of NF-κB p65 (red) was visualized using confocal microscopy after immunofluorescent staining. Nuclei were stained with DAPI (blue). Results are representative of those obtained from three independent tests.

**Figure 7 molecules-27-05037-f007:**
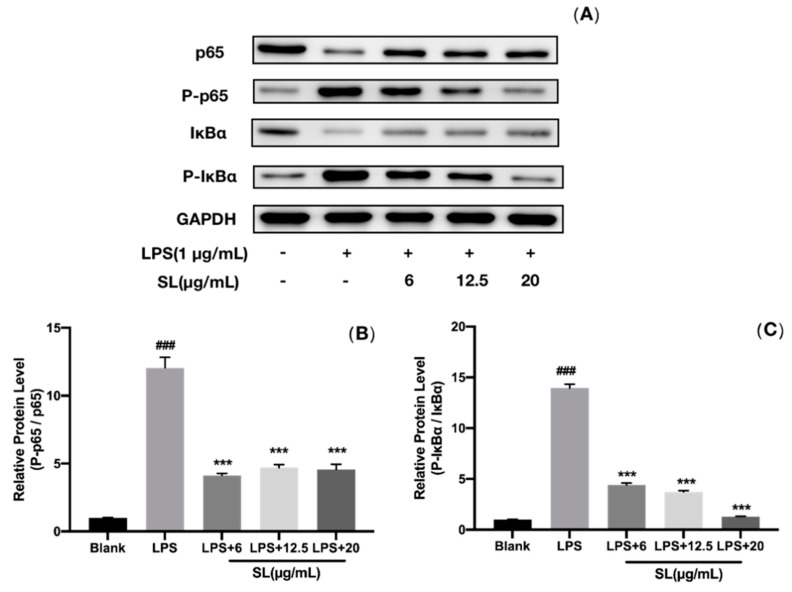
SLs inhibited the phosphorylation of NF-κB pathway-related protein in LPS-induced RAW264.7 cells. Cells were incubated with LPS (1 μg/mL) with or without SL for 18 h. (**A**) Western blot protein bands; (**B**,**C**) relative protein levels. The results are presented as the mean ± standard error (n = 3), and a *p*-value less than 0.05 was expressed as statistically significant (###, *p* < 0.001 vs. blank group; ***, *p* < 0.001 vs. LPS treatment group).

**Figure 8 molecules-27-05037-f008:**
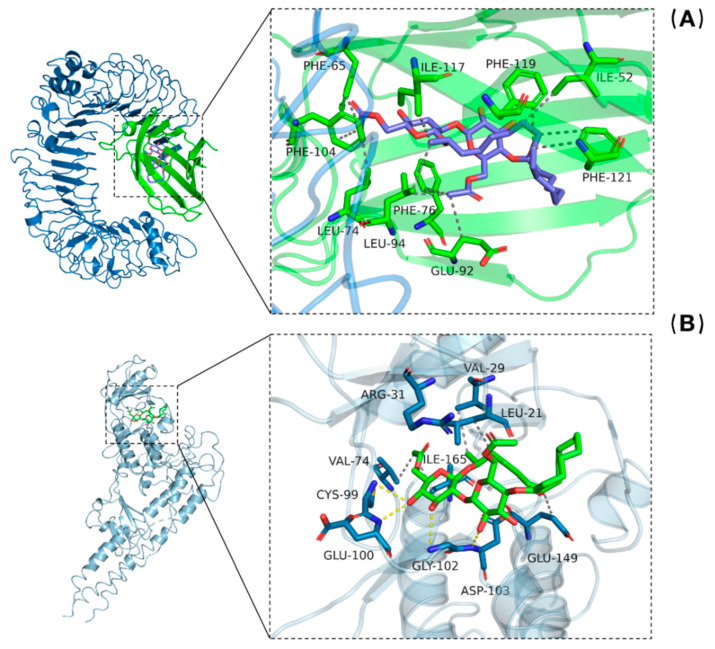
Molecular docking analysis of SL with (**A**) TLR4/MD-2 and (**B**) IKKβ. SL can bind to the TLR4/MD-2 protein complex with amino acids ILE, GLU, PHE, VAL, ALG, and LEU. SL interacts with IKKβ through amino acids CYS, GLU, ASP, GYL, ILE, GLU, PHE, VAL, ALG, and LEU.

**Figure 9 molecules-27-05037-f009:**
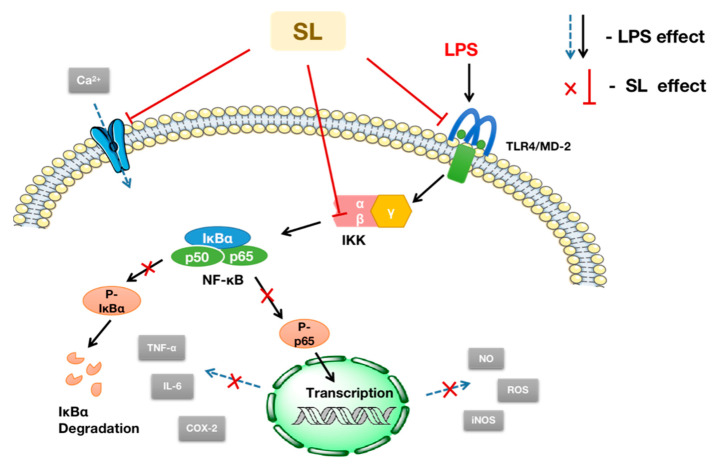
Schematic illustration of the anti-inflammatory action and mechanism of SL in a macrophage RAW264.7 Cells.

**Table 1 molecules-27-05037-t001:** Different Structural Modifications of SL.

Components	Formula (Acidic)	Mass	Percentage (%)	Formula (Lactonic)	Mass	Percentage (%)
Mona acetylation CH_3_	C_32_H_54_O_14_	662	10.84	C_32_H_56_O_13_	648	76.42
				C_32_H_54_O_13_	646	
Di acetylation CH_3_	C_34_H_60_O_15_	708	12.74	C_34_H_58_O_14_	690	
	C_34_H_58_O_15_	706		C_34_H_56_O_14_	688	
	C_34_H_56_O_15_	704		C_34_H_54_O_14_	686	

## Data Availability

The data used to support the findings of this study are available upon request from the corresponding author.

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
