# Peer review of "Sophorolipid Suppresses LPS-Induced Inflammation in RAW264.7 Cells through the NF-κB Signaling Pathway"

_molecules, 2022, doi:10.3390/molecules27155037_

Round 1
Reviewer 1 Report
Comments to the author
In the present manuscript, Xu et al. demonstrated that the characterization of sophorolipid and its anti-inflammatory effect in RAW264.7 cells. The authors demonstrated that sophorolipid exerted an anti-inflammatory activity via the pathway of NF-kB in RAW24.7 cells.
However, there are some points that need to be addressed by the authors.
Major concern:
l After confirming lactonic SL in SL characterization, why did the authors use SL in all subsequent experiments using RAW264.7 cell?
l In the Discussion part, line 327, authors described that ‘there is a dose-dependence in the inhibition of the TNF-a and IL-6 release’. However, IL-6 was similarly inhibited regardless of the SL concentration. Please describe these results and explain the authors’ opinion about the reason.
Minor concerns:
l In the Materials and methods part, line 117, there are no names of city and state of USA for Gibco. Please describe the name of city and state of USA for Gibco.
l In the Materials and methods part, line 135, ‘NO’ followed by ‘(TNF-a and IL-6)’. Please correct it.
l In the Y-axis label of Figure 5A and B, please correct from ‘Relative level of’ to ‘Relative mRNA expression level of’.
Reviewer 2 Report
The current manuscript deal with the study of the anti-inflammatory properties of sophorolipid on macrophages cell lines and the corresponding mechanism. Overall, the study is very well conducted and the results are convincing. However, the introduction focussing on the use of sophorolipid as bio-surfactant, albeit appealing, is not supported by the study. One would have expected at least some characterization and/demonstration of the interfacial activity of the compounds. For example what is the CMC of SF? Are the experiments carried out below or above that CMC? This will have an impact on the conformation/morphology of SF. Not much data is asked, but can SF produce an emulsion?
On the other hand, the production of SF is rather puzzling. With a purity of around 75 %, can the authors be sure that the main effect observed in the in vitro is due only to SF and not one of the “impurities?
Finally, the caption of all the figures must be more detailed and explain clearly what was studied and in which conditions. Also, please have a look at the English grammar and some spelling.
Round 2
Reviewer 2 Report
The authors have addressed correctly all the comments and the manucript is now recommended for publication.
This manuscript is a resubmission of an earlier submission. The following is a list of the peer review reports and author responses from that submission.